# Design, Synthesis, and Mechanistic Study of 2-Pyridone-Bearing Phenylalanine Derivatives as Novel HIV Capsid Modulators

**DOI:** 10.3390/molecules27217640

**Published:** 2022-11-07

**Authors:** Xujie Zhang, Lin Sun, Shujing Xu, Xiaoyu Shao, Ziyi Li, Dang Ding, Xiangyi Jiang, Shujie Zhao, Simon Cocklin, Erik De Clercq, Christophe Pannecouque, Alexej Dick, Xinyong Liu, Peng Zhan

**Affiliations:** 1Department of Medicinal Chemistry, Key Laboratory of Chemical Biology (Ministry of Education), School of Pharmaceutical Sciences, Shandong University, 44 West Culture Road, Jinan 250012, China; 2Department of Pharmacy, Qilu Hospital of Shandong University, 107 West Culture Road, Jinan 250012, China; 3Specifica, Inc., 1607 Alcaldesa Street, Santa Fe, NM 87501, USA; 4Rega Institute for Medical Research, Laboratory of Virology and Chemotherapy, K.U. Leuven, Herestraat 49 Postbus 1043 (09.A097), B-3000 Leuven, Belgium; 5Department of Biochemistry & Molecular Biology, Drexel University College of Medicine, Philadelphia, PA 19102, USA

**Keywords:** HIV, capsid, phenylalanine derivatives, protein-protein interaction

## Abstract

The AIDS pandemic is still of importance. HIV-1 and HIV-2 are the causative agents of this pandemic, and in the absence of a viable vaccine, drugs are continually required to provide quality of life for infected patients. The HIV capsid (CA) protein performs critical functions in the life cycle of HIV-1 and HIV-2, is broadly conserved across major strains and subtypes, and is underexploited. Therefore, it has become a therapeutic target of interest. Here, we report a novel series of 2-pyridone-bearing phenylalanine derivatives as HIV capsid modulators. Compound **FTC-2** is the most potent anti-HIV-1 compound in the new series of compounds, with acceptable cytotoxicity in MT-4 cells (selectivity index HIV-1 > 49.57; HIV-2 > 17.08). However, compound **TD-1a** has the lowest EC_50_ in the anti-HIV-2 assays (EC_50_ = 4.86 ± 1.71 μM; CC_50_
**=** 86.54 ± 29.24 μM). A water solubility test found that **TD-1a** showed a moderately increased water solubility compared with **PF74**, while the water solubility of **FTC-2** was improved hundreds of times. Furthermore, we use molecular simulation studies to provide insight into the molecular contacts between the new compounds and HIV CA. We also computationally predict drug-like properties and metabolic stability for **FTC-2** and **TD-1a**. Based on this analysis, **TD-1a** is predicted to have improved drug-like properties and metabolic stability over **PF74**. This study increases the repertoire of CA modulators and has important implications for developing anti-HIV agents with novel mechanisms, especially those that inhibit the often overlooked HIV-2.

## 1. Introduction

Human immunodeficiency virus (HIV), causing acquired immune deficiency syndrome (AIDS), remains one of the most serious global problems threatening human health [1]. HIV is comprised two main types, HIV type 1 (HIV-1) and HIV type 2 (HIV-2), which have similar modes of replication, transmission, and clinical symptomatology, but HIV-2 is less widespread and has reduced infectivity and transmissibility [2,3]. By simultaneously targeting different steps of the HIV life cycle, combined antiretroviral therapy (cART) has achieved remarkable success in reducing overall morbidity and mortality [4]. However, in the absence of a viable vaccine, taking these antiviral drugs is a life-long commitment. With the time people are required to remain on their regimens, drug resistance and side effects caused by cART represent real challenges [5]. One way to circumvent the problem of drug resistance is to develop new anti-HIV drugs with new targets and modes of action [6].

HIV capsid (CA) performs critical functions throughout the virus’s life cycle, making it an important pharmacological target [7,8]. The mature HIV-1 capsid is a lattice composed of ~ 1500 CA monomers which self-assemble into the typical fullerene cone structure composed of 12 pentamers and ~ 250 hexamers (Figure 1) [9,10]. Each CA monomer contains an *N*-terminal domain (NTD), a *C*-terminal domain (CTD), and a flexible linker connecting the two domains. During the formation of pentamers and hexamers, adjacent monomers are connected by NTD-NTD interactions and NTD-CTD interactions. In contrast, pairs of subunits between hexamers are connected by CTD-CTD interactions [11,12]. Both genetic and pharmacological studies have demonstrated that the stability of the capsid is fine-tuned such that any increase or decrease has deleterious effects on the replication of the virus [13].

Research on HIV capsid modulators has revealed several vulnerable regions of the CA protein structure [14,15,16,17,18,19,20]. Despite multiple pockets to which CA modulators could bind, particular attention has been given to a pocket formed between two adjacent protomers within the hexamer: the interprotomer pocket. This interprotomer pocket is broadly conserved across strains and the binding site for critical host cell factors such as cleavage and polyadenylation specific factor 6 (CPSF6) [21,22,23,24,25] and nucleoporins 153 and 358 (NUP153, NUP358) [23,24,26,27,28,29,30,31,32,33]. Despite the high conservation within the NTD-CTD, affinity alterations across clades have been observed for CPSF6 and NUP153 peptides targeting this site, and a complete loss of binding of NUP153 peptide (residue 1407 to 1423) to HIV-1 CA from clade A1 [34]. This points towards slight alterations of nuclear translocation mechanisms across clades. In addition, this essential interprotomer pocket is also targetable by small molecules, as evidenced by its being the binding site for the two most studied CA compounds; **PF-3450074** (**PF74**) and **GS-6207** (Figure 2A) [35]. **PF74** is a peptidomimetic reported by Pfizer [36], which is proven to inhibit HIV-1 replication with a bimodal mechanism. **PF74** stabilizes the CA core structure post-infection, thus inhibiting the uncoating process and HIV-1 reverse transcription in the early stage. In the late-stage, **PF74** destabilizes CA, causing aberrant virus morphologies that do not undergo maturation [20,35,36,37]. The co-crystal structure revealed the binding mode of **PF74**, in which the benzyl group of phenylalanine is essential for compound binding, and the indole is oriented towards the NTD of the adjacent subunit, forming a key cation-π interaction with Lys70 (Figure 2B) [38]. Even though the mechanism of **PF74** is novel, inferior metabolic stability precludes further studies [20]. **GS-6207** (Lenacapavir), by Gilead Science, is a compound derived from the heavy decoration of **PF74** and that shows robust antiviral activity in MT-4 cells in the low picomolar range (EC_50_ = 105 pM) [39]. **GS-6207** stabilizes the capsid, leading to a buildup of intact core in the cytoplasm, similar to **PF74**. Because it binds to the same pocket as **PF74** (Figure 2C), **GS-6207** prevents adequate binding of NUP153 and CPSF6 to the CA [40]. **GS-6207** has stable metabolic stability, allowing for once every six month injection therapy, which is vital to reducing toxicity, reducing patient stigma, and improving medication adherence [12]. This drug has been approved by the EC (European Community) and is the first drug to target CA available in the clinic.

Although **GS-6207** has many advantages, its synthetic strategy is complicated, and the cost is high, probably making it of consequence for patients in wealthy countries only. Moreover, drug-resistant strains have also already appeared in clinical trials [41]. Therefore, developing a new generation of drugs targeting CA that address these inadequacies is imperative.

To address this need, we proceeded to modify **PF74** and replace the indole with 2-pyridone. The rationale for this change is to form cation-π interactions with Lys70 and Arg173 on both sides, thereby increasing the affinity and potency of the compounds. Moreover, the carbonyl of 2-pyridone is expected to form hydrogen bonds with Lys70 and Arg173 of the adjacent subunit. The extended substituted phenyl groups are expected to form more abundant interactions with the amino acids of the adjacent subunit CTD, while phenyl groups and 2-pyridone are connected by an amide (Figure 3).

Herein, we report the design, synthesis, and mechanism study of 2-pyridone-bearing phenylalanine derivatives as novel HIV capsid modulators. The antiviral activities of all synthesized compounds were tested in MT-4 cells infected by HIV-1III_B_ or HIV-2 ROD, and structure-activity relationships were established. We performed docking studies to gain insights into the binding modes of representative compounds and to understand the biological potencies. Finally, the drug-like properties and metabolic stability of representative compounds and **PF74** were computationally predicted.

## 2. Chemistry

As shown in Figure 1, starting from commercially available 4-methoxy-*N*-methylaniline (**1**), the target compounds were prepared via a concise and well-established synthetic route, as outlined below. Treating of **1** with Boc-*L*-phenylalanine or Boc-3,5-difluoro-*L*-phenylalanine and 2-(7-Azabenzotriazol-1-yl)-*N*,*N*,*N*’,*N*’-tetramethyluronium hexafluorophosphate (HATU) in *N*,*N*-diisopropylethylamine (DIEA) and dichloromethane afforded **2a** or **2b**, followed by removal of *tert*-butyloxycarbonyl (Boc) protection resulted in the formation of free amine **3a** or **3b**. The intermediate **4a** or **4b** was obtained by acylation of **3a** or **3b** with bromoacetic acid in dichloromethane solution. The nucleophilic substitution of 2-pyridone with **4a** or **4b** in THF resulted in **TC-1** or **FTC-1**. We then removed the Boc group and obtained **TC-2** or **FTC-2**. Finally, triethylamine (TEA), **TC-2** or **FTC-2** were added, and benzoyl chloride was substituted by different groups in dichloromethane to afford the compounds **TD-1a-1l**. Another target compound, **TD-1m**, was prepared by a hydrogenation reduction of the nitro group of **TD-1l**.

## 3. Results and Discussion

### 3.1. In Vitro Anti-HIV Assays and SAR Analysis for HIV-1 Potency

The antiviral activities and cytotoxicities of the target compounds were tested using MT-4 cells infected by HIV-1 III_B_ or HIV-2 ROD. Table 1 shows the EC_50_ and CC_50_ values of each compound. **PF74** was utilized as the control drug in this assay.

The majority of the newly designed compounds showed anti-HIV activities in MT-4 cells, in which **FTC-2** exhibited the best anti-HIV-1 activity with an EC_50_ value of 5.36 ± 0.98 μM. Meanwhile, **TD-1a** was the most potent compound inhibiting HIV-2 replication (EC_50_ = 4.86 ± 1.71 μM), surpassing the parental compound **PF74**. In addition, both compounds showed acceptable cytotoxicity in MT-4 cells (**FTC-2** selectivity index HIV-1 > 49.57; HIV-2 > 17.08, **TD-1a** selectivity index HIV-1 = 5.15; HIV-2 = 17.81).

As shown in Table 1, fluorine substituting R_1_ enhanced the anti-HIV-1 activity (**FTC-2** > **TC-2**). The 2-pyridone bearing side chain did not show the expected activity increase for the compounds when testing the anti-HIV-1 activity. In contrast, modifying the amino group of **TC-2** did result in potency improvement for most compounds (**TD-1a**, **TD-1c**, **TD-1d**, **TD-1g**, **TD-1i**, **TD-1j**) over the parental **TC-2**. Nevertheless, different substituents of R_2_ have different effects on the anti-HIV-1 activity, although the activities of most substituted benzene compounds were reduced. When R_2_ is carbonyl benzene substituted by fluorine, the *ortho*-substitution was superior to the *meta*- and *para*-substitution, where the *para*-substitution lost its anti-HIV-1 potency (**TD-1d** > **TD-1c** > **TD-1b**). The law of loss of effectiveness for *para*-substitution also applies to some other weak electron-withdrawing substituents, such as chlorine, bromine, and the nitro group (**TD-1e**, **TD-1f**, **TD-1l**). However, when the *para*-position was substituted with strong electron-withdrawing groups (**TD-1i**, **TD-1j**), the compounds showed comparable or slightly weakened activity than the unsubstituted counterparts. In conclusion, for this series of compounds, the substitution of R_1_ by fluorine and the substitution of R_2_ by a robust electron-withdrawing group in the *para*-position substituting or unsubstituted phenylcarbonyl group were beneficial in enhancing the anti-HIV-1 activity.

For anti-HIV-2 efficacy, this series of compounds showed enhanced activity compared to **PF74**. Fluorine substituting R_1_ enhanced anti-HIV-2 activity (**FTC-2** > **TC-2**), and modifying the amino group substantially improved potency. Another aspect of the anti-HIV-1 experiment was that the compound was the most potent when R_2_ was substituted with an unsubstituted phenylcarbonyl group (**TD-1a**). When the R_2_-benzene of **TD-1a** was replaced by fluorine, the activities were reduced. However, unlike in the anti-HIV-1 experiment, the unsubstituted compounds did not lose their activity, and the activity laws differed (*meta* > *ortho* > *para*). When the electron-donating group is substituted for the *para*-position, the activity decreases to a small extent compared to the electron-withdrawing group. In conclusion, compared to targeting HIV-1, this series of compounds has a stronger ability to inhibit HIV-2 replication that is superior to **PF74**.

### 3.2. Solubility of Representative Compounds

To test the changes in water solubility when two more amides or its mimics were introduced to the target compounds, which may lower the solubility of the target compounds, we performed water solubility assays (Table 2). In the HPLC assay, the peak area of PF74 saturated aqueous solution under pH 2 is much smaller than the minimum value of its standard curve (7.03 mAu, 1.12 μg/mL). Therefore, we cannot calculate its water solubility. It was illustrated that the solubility of compounds varied greatly under different pH. **TD-1a** had a moderately increased water solubility compared with **PF74**, while the water solubility of **FTC-2** was improved by a factor of hundreds, which may be due to the introduction of its arylamine group.

### 3.3. Binding Mode of **FTC-2** within the Interprotomer Pocket

To shed light on the binding mode of representative compounds to the HIV-1 capsid, molecular docking of the most potent compound, **FTC-2**, was performed using Schrödinger 2022-1 software (Schrödinger, New York, NY, USA).

As shown in Figure 4, **FTC-2** had two main binding modes in the docking study, in which the difference was mainly the orientation of the 2-pyridone moiety. When the carbonyl group of 2-pyridone faced the inside of the binding pocket, its carbonyl formed hydrogen bonds with Arg173 and Lys70, but the original cation-π interactions of Lys70 and Arg173 were absent (Figure 4A). On the contrary, if the carbonyl group of the 2-pyridone moiety is kinked “upwards” towards the solvent, it results in the formation of cation-π interactions sandwiched by Lys70 and Arg173 (Figure 4B). In addition, both conformations formed a hydrogen bond with the main chain carbonyl of Tyr169. Although there were two binding modes and the glide scores of the two modes were almost the same, the interaction of **FTC-2** with the critical amino acid Lys70 was weakened compared to **PF74**, and the hydrogen bond with Gln63 was also lost, which may be one of the reasons for the decreased potency.

### 3.4. Computational Assessment of Drug-like Properties and Metabolic Stability

**PF74** suffers from several problems that limit its clinical use that are primarily related to its drug-like properties. Therefore, we sought to analyze the new compounds’ predicted ADME properties (absorption, distribution, metabolism, and excretion) and compare them with **PF74** (Figure 5). To accomplish this comparison, we used in silico prediction of drug-like metrics of the results as implemented in the oral non–central nervous system (CNS) drug profile in StarDrop 7 (Optibrium, Ltd., Cambridge, UK) [42]. This profile consists of several models, weighted differently and combined into a single score by a probabilistic scoring algorithm. For reference, scores range from 0 to 1, with 0 suggesting extremely non–drug-like and 1 suggesting the perfect drug.

Optibrium’s oral non-CNS drug profile is composed of the following metrics: logS (intrinsic aqueous solubility); classification for human intestinal absorption; logP (octanol/water); hERG (human ether-à-go-go-related gene) pIC50 (mammalian cells); cytochrome P450 CYP2D6 classification; cytochrome P450 CYP2C9 pKi values; classification of P-glycoprotein transport; classification of blood-brain barrier (BBB) penetration; and predicted BBB penetration value. The models and their respective importance to the profile are shown in Figure 5A. As can be seen, **FTC-2** and **TD-1a** display a four- and five-fold improved score compared to **PF74**. **TD-1a** has also improved aqueous solubility compared to **PF74**, as judged by the logS, which improves overall bioavailability. Moreover, **PF74** and **FTC-2** are not a substrate of the P-Glycoprotein (P-gp), and **TD-1a** is predicted to be a substrate; however, one with a lower probability (Figure 5B). Overall, **FTC-2** and **TD-1a** show improved oral non-CNS drug profile scores, primarily due to improved solubility (logS and logD) and lower probability for plasma protein binding (Figure 5C), which increases bioavailability and is predicted to be absorbed by the human intestine (Figure 5D).

A significant hurdle for **PF74**′s development is its poor metabolic stability [20]. Orally administered drugs must first pass the intestinal wall, followed by the portal circulation to the liver before reaching the bloodstream. Both sides are locations for first-pass metabolism and can adversely metabolize drugs before adequate plasma concentrations are reached. Therefore, we next sought to investigate whether or not our compounds had improved predicted metabolic stability. We employed a computational analysis first demonstrated to be an accurate indicator of metabolic stability by the Cocklin group [19,43,44]. This analysis uses the P450 module in StarDrop V7 (Optibrium, Ltd., Cambridge, UK) to predict each compound’s major metabolizing Cytochrome P450 isoforms using the WhichP450™ model, followed by affinity prediction to that isoform using the HYDE function in SeeSAR (BioSolveIT Gmbh, Sankt Augustin, Germany) [44,45,46]. The results of this analysis are shown in Figure 6.

All compounds, including **PF74**, are predicted to be primarily metabolized by the CYP3A4 isoform (Figure 6A). Moreover, and in agreement with our computational prediction, a recent study also experimentally verified that **PF74s** poor metabolic stability is attributed mainly to its poor resistance to cytochrome P450 3A4 [18]. We therefore, analyzed the predicted metabolic lability of our compounds and **PF74** by the CY3A4 isoform by comparing the overall composite site lability (CSL) score and the number of labile sites. The CSL score reflects the overall efficiency of metabolism of the molecule by combining the labilities of individual sites within the compound. The number of labile sites for **TD-1a** is comparable to **PF74,** while **FTC-2** displays three labile sites primarily at the secondary amine, the hydroxyl group, and the carbon between the two electronegative fluorine atoms, which apply an electron-withdrawing effect on this carbon atom.

The slight nuances in the CSL score between those three indicated increased metabolic stability in the following order **FTC-2, TD-1a,** and **PF74** (Figure 6B), with **PF74** displaying the lowest CSL score and thereby indicating higher metabolic stability.

In addition to the CSL score and the number of labile sites, which assume that all compounds bind with similar affinity to the CYP3A4 isoform, other factors such as compound reduction rate and actual binding affinity to the CYP3A4 isoform can infer metabolic stability. Moreover, binding affinity can also be influenced by intrinsic compound properties such as size and lipophilicity. We, therefore, performed predictive binding affinity calculations using the HYdrogen bond and DEhydration (HYDE) energy scoring function in SeeSAR 12.1 (BioSolveIT Gmbh, Sankt Augustin, Germany). For this analysis, we used the structure of the human CYPA4 bound to an inhibitor (PDB ID 4D78). The HYDE scoring function in SeeSAR provides a range of affinities, spanning an upper and lower limit. Therefore, we used the lower limit as the affinity parameter to compare **FTC-2**, **TD-1a**, and **PF74** (Figure 6C), which indicated low nM-affinity for **PF74** to CYP3A4, while **FTC-2** and especially **TD-1a** have significantly lower predicted CYP3A4 affinity. Combining the results from these predictions (CSL scores, labile sites, and predicted CYP3A4 affinity), this analysis indicates that compound **TD-1a** should have improved metabolic stability than **PF74**, primarily due to the significantly lower CYP3A4 affinity.

Next, we evaluated potential toxicities associated with our compounds to identify possible issues and address them early in our drug development pipeline. We, therefore, include genotoxicity and hepatotoxicity endpoints in our multiparameter optimization for **FTC-2**, **TD-1a**, and **PF74** using the Derek Nexus module within Stardrop V7. Derek Nexus is a knowledge- and rule-based expert system for semi-quantitatively estimating DNA reactive moieties within molecules. Based on this prediction, none of our compounds, including **PF74**, show any concerning likelihood of genotoxicity or hepatotoxicity (Figure 7). In contrast, and used as positive controls in our prediction, ethyl methanesulfonate (EMS) and Lumiracoxib are known to have in vivo genotoxic and hepatotoxic issues.

## 4. Conclusions

In this study, a novel series of 2-pyridone-bearing phenylalanine derivatives have been designed, synthesized, and evaluated as HIV capsid modulators. Most of the newly synthesized compounds displayed anti-HIV activity, in which **FTC-2** is the most potent anti-HIV-1 compound (EC_50_ = 5.36 ± 0.98 μM), while **TD-1a** is more potent at blocking the replication of HIV-2 (EC_50_ = 4.86 ± 1.71 μM). Moreover, both compounds showed acceptable cytotoxicity (**FTC-2** selectivity index HIV-1 > 49.57; HIV-2 > 17.08, **TD-1a** selectivity index HIV-1 = 5.15; HIV-2 = 17.81), comparable to the control compound **PF74**. The water solubility test found that **TD-1a** showed a small increase or equivalent water solubility compared with PF74, while the water solubility of **FTC-2** was improved by a factor of hundreds. Although molecular docking showed that **FTC-2** had an ideal binding mode, some key functions were still missing, causing a decrease in antiviral potency. Notably, the orientation of the carbonyl on the 2-pyridone moiety could contribute to a lower potency by orienting the ring unfavorably for cation-π interactions by Lys70 and Arg173. Future modifications of this series could account for this. For example, stronger electron-donating groups, such as alkyl groups, could increase electronegativity within the ring system and the probability of cation-π interactions for potency improvement.

The water solubility test found that **TD-1a** showed a moderately increased water solubility compared with PF74, while the water solubility of **FTC-2** was improved by a factor of hundreds. Furthermore, using a computational approach, we predicted drug-like properties and metabolic stability for **FTC-2** and **TD-1a**, of which **TD-1a** showed an improved overall drug-likeness score and metabolic stability. These predictions, although previously demonstrated as indicators of drug-likeness and metabolic stability, have to be experimentally verified in future studies. Although the anti-HIV-1 activity of the compounds was reduced, the anti-HIV-2 activity of **TD-1a** was improved relative to **PF74** and **TC-2** (a non-fluorine-substituted **FTC-2**). This result shows that the modification of the terminal amino group of **TC-2** is tolerable. We believe this study positively contributes to the repertoire of novel inhibitor chemotypes targeting the HIV capsid protein and increases the likelihood of advancing to synthetically accessible, potent antivirals.

## 5. Experimental Section

### 5.1. Chemistry

^1^H NMR and ^13^C NMR spectra were recorded on a Bruker AV-400 spectrometer or Bruker AV-600 spectrometer (Bruker, Billerica, MA, USA) using solvents as indicated (DMSO-*d_6_*). Chemical shifts were reported in *δ* values (ppm) with tetramethylsilane (TMS) as the internal reference, and *J* values were reported in hertz (Hz). Melting points (mp) were determined on a micromelting point apparatus (Tianjin Optics, Tianjin, China) and were uncorrected. TLC was performed on Silica Gel GF254 for TLC (Merck KGaA, Darmstadt, Germany), and spots were visualized by iodine vapor or irradiation with UV light (*λ* = 254 nm). Flash column chromatography was performed on a column packed with Silica Gel60 (200–300 mesh). Thin-layer chromatography was performed on pre-coated HUANGHAI_HSGF254 (Yantai, Shandong, China), 0.15–0.2 mm TLC-plates. Solvents were of reagent grade and were purified and dried by standard methods when necessary. The concentration of the reaction solutions involved the use of a rotary evaporator at reduced pressure. The solvents of dichloromethane, TEA and methanol etc., were obtained from Sinopharm Chemical Reagent Co., Ltd. (SCRC, Shanghai, China), and were of AR grade. The key reactants, including 4-methoxy-*N*-methylaniline, *N*-(*tert*-butoxycarbonyl)-*L*-phenylalanine etc. were purchased from Bide Pharmatech Co., Ltd. (Shanghai, China) or Shanghai Haohong Scientific Co., Ltd. (Shanghai, China) The purity of the final representative compounds was checked by HPLC and was >95%.

#### 5.1.1. General Procedure for the Synthesis of **2a** and **2b**

To a solution of Boc-*L*-phenylalanine or Boc-3,5-difluoro-*L*-phenylalanine (1 eq.) in 20 mL dichloromethane, HATU (1.5 eq.) was added at 0 °C, and the mixture was stirred for 0.5 h. Subsequently, DIEA (2 eq.) and 4-methoxy-*N*-methylaniline (1.2 eq.) were added to the mixture and then stirred at room temperature for another 2 h (monitored by TLC). The reaction solution was initially washed with saturated sodium bicarbonate, extracted with DCM (3 × 20 mL), dried over anhydrous Na_2_SO_4_, filtered, and concentrated under reduced pressure to afford a corresponding crude product, purified by flash column chromatography to afford intermediates **2a** and **2b**.

##### tert-butyl (S)-(1-((4-methoxyphenyl)(methyl)amino)-1-oxo-3-phenylpropan-2-yl)carbamate (**2a**)

Yellow oil, yield: 81%. ^1^H NMR (400 MHz, DMSO-*d*_6_): *δ* 7.22 (d, *J* = 8.3 Hz, 2H, Ph-H), 7.20–7.11 (m, 3H, Ph-H), 7.09 (d, *J* = 8.2 Hz, 1H, Ph-H), 7.03 (d, *J* = 8.6 Hz, 2H, Ph-H), 6.79 (d, *J* = 7.3 Hz, 2H, Ph-H), 4.27–4.06 (m, 1H, CH), 3.81 (s, 3H, OCH_3_), 3.13 (s, 3H, NCH_3_), 2.75 (dd, *J* = 13.4, 3.8 Hz, 1H, PhCH), 2.61 (dd, *J* = 13.3, 10.3 Hz, 1H, PhCH), 1.30 (s, 9H, C(CH_3_)). ESI-MS: m/z 385.4 (M + 1)^+^, 407.5 (M + 23)^+^. C_22_H_28_N_2_O_4_ [384.48].

##### tert-butyl (S)-(3-(3,5-difluorophenyl)-1-((4-methoxyphenyl) (methyl)amino)-1-oxopropan-2-yl)carbamate (**2b**)

Yellow oil, yield: 89%. ^1^H NMR (400 MHz, DMSO-*d*_6_): *δ* 7.31 (d, *J* = 8.4 Hz, 2H, Ph-H), 7.13–7.04 (m, 3H, Ph-H), 7.01 (d, *J* = 9.5 Hz, 1H, Ph-H), 6.44 (d, *J* = 8.3 Hz, 2H, Ph-H), 4.20–4.10 (m, 1H, CH), 3.81 (s, 3H, OCH_3_), 3.14 (s, 3H, NCH_3_), 2.82–2.61 (m, 2H, PhCH_2_), 1.29 (s, 9H, C(CH_3_)). ESI-MS: m/z 421.07 (M + 1)^+^, 443.17 (M + 23)^+^. C_22_H_26_F_2_N_2_O_4_ [420.46].

#### 5.1.2. General Procedure for the Synthesis of **3a** and **3b**

Trifluoroacetic acid (5.0 eq.) was added dropwise to the corresponding substituted intermediate **2a** or **2b** (1.0 eq.) in 30 mL dichloromethane and stirred at room temperature for 1 h (monitored by TLC). The resulting mixture solution was then alkalized to pH ~7 with saturated sodium bicarbonate solution and then extracted with dichloromethane (3 × 30 mL), dried over anhydrous Na_2_SO_4_, filtered, and concentrated under reduced pressure to afford corresponding crude products **3a** and **3b**.

##### (*S*)-2-amino-N-(4-methoxyphenyl)-N-methyl-3-phenylpropanamide (**3a**)

Yellow oil, yield: 75%. ^1^H NMR (400 MHz, DMSO-*d*_6_): *δ* 7.29–7.13 (m, 3H, Ph-H), 7.03–6.75 (m, 6H, Ph-H), 3.77 (s, 3H, OCH_3_), 3.44–3.35 (m, 1H, CH), 3.06 (s, 3H, NCH_3_), 2.75 (dd, *J* = 12.8, 6.7 Hz, 1H, PhCH), 2.45 (dd, *J* = 12.9, 7.1 Hz, 1H, PhCH), 1.87 (s, 2H, NH_2_). ESI-MS: m/z 285.05 (M + 1)^+^. C_17_H_20_N_2_O_2_ [284.36].

##### (*S)*-2-amino-3-(3,5-difluorophenyl)-N-(4-methoxyphenyl)-N-methylpropanamide (**3b**)

Yellow oil, yield: 77%. ^1^H NMR (400 MHz, DMSO-*d*_6_): *δ* 7.10–6.93 (m, 5H, Ph-H), 6.57 (h, *J* = 4.1 Hz, 2H, Ph-H), 3.78 (s, 3H, OCH_3_), 3.35 (dd, *J* = 7.6, 5.9 Hz, 1H, CH), 3.09 (s, 3H, NCH_3_), 2.74 (dd, *J* = 13.1, 5.8 Hz, 1H, PhCH), 2.54–2.45 (m, 1H, PhCH), 1.82 (s, 2H, NH_2_). ESI-MS: m/z 321.11 (M + 1)^+^, m/z 343.25 (M + 23)^+^. C_17_H_18_F_2_N_2_O_2_ [320.34].

#### 5.1.3. General Procedure for the Synthesis of **4a** and **4b**

Bromoacetic acid (1.2 eq.) and HATU (1.5 eq.) were mixed in 15 mL dichloromethane and stirred in an ice bath for 0.5 h. The corresponding substituted intermediate **3a** or **3b** (1 eq.) and DIEA (2 eq.) were then slowly added to the above solution at 0 °C. The reaction system was then stirred at room temperature for an additional 0.5 h (monitored by TLC). The resulting mixture was initially washed with saturated sodium bicarbonate and extracted with DCM (3 × 20 mL) and dried over anhydrous Na_2_SO_4_, filtered, and concentrated under reduced pressure to afford a corresponding crude product, purified by flash column chromatography to afford intermediates **4a** and **4b**.

##### (*S)*-2-(2-bromoacetamido)-N-(4-methoxyphenyl)-N-methyl-3-phenylpropanamide (**4a**)

White oil, yield: 73%. ^1^H NMR (600 MHz, DMSO-*d*_6_): *δ* 8.62 (d, *J* = 7.9 Hz, 1H, NH), 7.22–7.16 (m, 3H, Ph-H), 7.05 (d, *J* = 8.3 Hz, 2H, Ph-H), 6.96 (d, *J* = 9.2 Hz, 2H, Ph-H), 6.88 (d, *J* = 6.2 Hz, 2H, Ph-H), 4.44 (td, *J* = 8.4, 5.6 Hz, 1H, CH), 3.82 (d, *J* = 2.4 Hz, 2H, CH_2_), 3.79 (s, 3H, OCH_3_), 3.10 (s, 3H, NCH_3_), 2.87 (dd, *J* = 13.5, 5.5 Hz, 1H, PhCH), 2.65 (dd, *J* = 13.5, 8.7 Hz, 1H, PhCH). ESI-MS: m/z 405.4 (M + 1)^+^. C_19_H_21_BrN_2_O_3_ [405.29].

##### (*S*)-2-(2-bromoacetamido)-3-(3,5-difluorophenyl)-N-(4-methoxyphenyl)-N-methylpropanamide (**4b**)

White solid, yield: 78%. ^1^H NMR (400 MHz, DMSO-*d*_6_): *δ* 8.70 (d, *J* = 8.0 Hz, 1H, NH), 7.22 (d, *J* = 8.8 Hz, 2H, Ph-H), 7.04 (d, *J* = 8.6 Hz, 3H, Ph-H), 6.52 (d, *J* = 6.3 Hz, 2H, Ph-H), 4.44 (qd, *J* = 8.6, 4.6 Hz, 1H, CH), 3.81 (s, 2H, BrCH_2_), 3.80 (s, 3H, OCH_3_), 3.13 (s, 3H, NCH_3_), 2.89 (dd, *J* = 13.7, 4.6 Hz, 1H, PhCH), 2.69 (dd, *J* = 13.7, 9.3 Hz, 1H, PhCH). ESI-MS: m/z 443.15 (M + 2)^+^, m/z 463.16 (M – 1 + 23)^+^. C_17_H_18_F_2_N_2_O_2_ [441.27].

#### 5.1.4. General Procedure for the Synthesis of **FTC-1** and **TC-1**

Under ice bath, *tert*-butyl (2-oxo-1,2-dihydropyridin-4-yl)carbamate (1.2 eq.) was dissolved in the solution of tetrahydrofuran (6 mL), and then the NaH (2 eq) was added slowly. The resulting mixture was then stirred for 30 min. The corresponding intermediate **4a** or **4b** was then added, and the resulting mixture was then stirred for 2 h at room temperature (monitored by TLC). The reaction mixture was then concentrated under reduced pressure, and the remains were resoluted with ethyl acetate. The resulting mixture was initially washed with saturated sodium bicarbonate, extracted with ethyl acetate (3 × 20 mL), dried over anhydrous Na_2_SO_4_, filtered, and concentrated under reduced pressure to give the corresponding crude product, which was purified by flash column chromatography to result in the products **FTC-1** and **TC-1**.

##### *tert*-butyl *(S)*-(1-(2-((3-(3,5-difluorophenyl)-1-((4-methoxyphenyl)(methyl)amino)-1-oxopropan-2-yl)amino)-2-oxoethyl)-2-oxo-1,2-dihydropyridin-4-yl)carbamate (**FTC-1**)

Yellow solid, yield: 83%.

##### *tert*-butyl *(S)*-(1-(2-((1-((4-methoxyphenyl)(methyl)amino)-1-oxo-3-phenylpropan-2-yl)amino)-2-oxoethyl)-2-oxo-1,2-dihydropyridin-4-yl)carbamate (**TC-1**)

Yellow solid, yield: 74%. ^1^H NMR (600 MHz, DMSO-*d*_6_) δ 9.55 (s, 1H, NH), 8.51 (d, *J* = 7.8 Hz, 1H, NH), 7.29 (d, *J* = 7.5 Hz, 1H, 2-Pyridone), 7.20 (d, *J* = 7.2 Hz, 2H, Ph), 7.00 (s, 2H, Ph), 6.90 (m, 5H, Ph), 6.47 (d, *J* = 2.3 Hz, 1H, 2-Pyridone), 6.29–6.26 (m, 1H, 2-Pyridone), 4.48 (d, *J* = 15.8 Hz, 1H, CH_2_), 4.42 (q, *J* = 8.0 Hz, 1H, CH), 4.35 (d, *J* = 15.8 Hz, 1H, CH_2_), 3.76 (s, 3H, OCH_3_), 3.08 (s, 3H, NCH_3_), 2.87 (dd, *J* = 13.5, 5.8 Hz, 1H, PhCH), 2.66 (dd, *J* = 13.5, 8.4 Hz, 1H, PhCH), 1.47 (s, 9H, Boc). ^13^C NMR (150 MHz, DMSO-*d*_6_) δ 171.30 (C=O), 167.19 (C=O), 162.43 (C=O), 158.96 (C=O), 152.63, 149.71, 140.13, 137.78, 135.93, 129.40 (2 × C), 129.05 (2 × C), 128.62 (2 × C), 126.91, 115.06 (2 × C), 101.88, 98.94, 80.69, 55.85, 51.86, 49.91, 38.09, 37.74, 28.41 (3 × C). ESI-MS: m/z 535.4 (M + 1)^+^. C_29_H_34_N_4_O_6_ [534.25].

#### 5.1.5. General Procedure for the Synthesis of **FTC-2** and **TC-2**

Trifluoroacetic acid (5.0 eq.) was added dropwise to the corresponding substituted intermediate **FTC-1** or **TC-1** (1.0 eq.) in 30 mL dichloromethane and stirred at room temperature for 1 h (monitored by TLC). The resulting mixture solution was then alkalized to pH ~7 with saturated sodium bicarbonate solution and then extracted with dichloromethane (3 × 30 mL), dried over anhydrous Na_2_SO_4_, filtered, and concentrated under reduced pressure to afford corresponding crude products, which was purified by flash column chromatography to afford products **FTC-2** and **TC-2**. (MS, ^1^H NMR and ^13^C NMR spectra are shown in Appendix A)

##### (*S)*-2-(2-(4-amino-2-oxopyridin-1(2H)-yl)acetamido)-3-(3,5-difluorophenyl)-N-(4-methoxyphenyl)-N-methylpropanamide (**FTC-2**)

Yellow solid, yield: 86%. m.p.: 63–65 °C. ^1^H NMR (600 MHz, DMSO-*d*_6_) δ 8.44 (d, *J* = 7.8 Hz, 1H, NH), 7.18 (d, *J* = 8.3 Hz, 2H, Ph), 7.06 (d, *J* = 7.4 Hz, 1H, 2-Pyridone), 7.02 (tt, *J* = 9.3, 2.5 Hz, 1H, Ph), 6.99–6.95 (m, 2H, Ph), 6.57–6.51 (m, 2H, Ph), 5.97 (d, *J* = 5.1 Hz, 2H, NH_2_), 5.62 (dd, *J* = 7.4, 2.4 Hz, 1H, 2-Pyridone), 5.21 (d, *J* = 2.5 Hz, 1H, 2-Pyridone), 4.42 (td, *J* = 8.1, 4.9 Hz, 1H, CH), 4.39–4.23 (m, 2H, CH_2_), 3.78 (s, 3H, OCH_3_), 3.11 (s, 3H, NCH_3_), 2.86 (dd, *J* = 13.7, 4.9 Hz, 1H, PhCH), 2.69 (dd, *J* = 13.7, 8.8 Hz, 1H, PhCH). ^13^C NMR (150 MHz, DMSO-*d*_6_) δ 170.86 (C=O), 168.03 (C=O), 162.68 (C=O), 162.55 (dd, ^1^*J_CF_* = 245.8, ^3^*J_CF_* = 13.3 Hz, 2 × C), 159.12, 157.72, 142.41 (t, ^3^*J_CF_* = 9.3 Hz), 139.52, 135.84, 129.10 (2 × C), 115.21 (2 × C), 112.42 (dd, ^2^*J_CF_* = 19.8, ^4^*J_CF_* = 4.9 Hz, 2 × C), 102.47 (t, ^2^*J_CF_* = 25.5 Hz), 98.71, 92.91, 55.89, 51.42, 49.48, 37.71, 37.38. ESI-MS: m/z 471.4 (M + 1)^+^, 493.5 (M + 23)^+^. C_24_H_24_F_2_N_4_O_4_ [470.18].

##### (*S)*-2-(2-(4-amino-2-oxopyridin-1(2H)-yl)acetamido)-N-(4-methoxyphenyl)-N-methyl-3-phenylpropanamide (**TC-2**)

Yellow solid, yield: 79%. m.p.: 164–168 ℃. ^1^H NMR (600 MHz, DMSO-*d*_6_) δ 8.39 (d, *J* = 7.7 Hz, 1H, NH), 7.25–7.13 (m, 4H, Ph, NH_2_), 7.03 (d, *J* = 7.4 Hz, 2H, Ph), 6.93–6.86 (m, 5H, Ph), 5.96 (d, *J* = 5.0 Hz, 1H, 2-Pyridone), 5.61 (dd, *J* = 7.4, 2.4 Hz, 1H, 2-Pyridone), 5.21 (d, *J* = 2.4 Hz, 1H, 2-Pyridone), 4.42 (td, *J* = 7.7, 5.6 Hz, 1H, CH), 4.30 (dd, *J* = 86.7, 15.8 Hz, 2H, CH_2_), 3.76 (s, 3H, OCH_3_), 3.08 (s, 3H, NCH_3_), 2.85 (dd, *J* = 13.5, 5.7 Hz, 1H, PhCH), 2.64 (dd, *J* = 13.5, 8.5 Hz, 1H, PhCH). ^13^C NMR (150 MHz, DMSO-*d*_6_) δ 171.34 (C=O), 167.89 (C=O), 162.70 (C=O), 158.96, 157.70, 139.57, 137.81, 135.96, 129.41 (2 × C), 129.07 (2 × C), 128.61 (2 × C), 126.89, 115.07 (2 × C), 98.68, 92.93, 55.86, 51.78, 49.44, 38.11, 37.74.ESI-MS: m/z 435.5 (M + 1)^+^. C_24_H_36_N_4_O_4_ [434.20].

#### 5.1.6. General Procedure for the Synthesis of **TD-1a–1l**

Under an ice bath, the key intermediate **TC-2** (1 eq.), corresponding substituted benzoyl chloride (1.5 eq.), TEA (2 eq.) were dissolved in the solution of dichloromethane (10 mL). The resulting mixture was then stirred at room temperature (monitored by TLC). The reaction mixture was then extracted with dichloromethane (20 mL), and the combined organic phase was washed with saturated NaCl solution (3 × 20 mL), dried over anhydrous Na_2_SO_4_, filtered, and concentrated under reduced pressure to give the corresponding crude product, which was purified by recrystallization or preparation thin layer chromatography to result in **TD-1a**–**1l**.

##### (*S*)-N-(1-(2-((1-((4-methoxyphenyl)(methyl)amino)-1-oxo-3-phenylpropan-2-yl)amino)-2-oxoethyl)-2-oxo-1,2-dihydropyridin-4-yl)benzamide (**TD-1a**)

Yellow solid, yield: 87%. m.p.: 102–107 ℃. ^1^H NMR (400 MHz, DMSO-*d*_6_) δ 10.31 (s, 1H, NH), 8.60 (d, *J* = 7.7 Hz, 1H, NH), 7.92 (d, *J* = 7.2 Hz, 2H, Ph), 7.63 (t, *J* = 7.2 Hz, 1H, Ph), 7.55 (t, *J* = 7.5 Hz, 2H, Ph), 7.40 (d, *J* = 7.5 Hz, 1H, 2-Pyridone), 7.21 (d, *J* = 6.9 Hz, 3H, Ph (2H), 2-Pyridone (1H)), 7.07–6.98 (m, 2H, Ph), 6.95–6.87 (m, 5H, Ph), 6.58 (dd, *J* = 7.5, 2.4 Hz, 1H, 2-Pyridone), 4.54 (d, *J* = 15.8 Hz, 1H, CH_2_), 4.44 (d, *J* = 8.1 Hz, 1H, CH), 4.40 (d, *J* = 15.9 Hz, 1H, CH_2_), 3.76 (s, 3H, OCH_3_), 3.09 (s, 3H, NCH_3_), 2.88 (dd, *J* = 13.6, 5.5 Hz, 1H, PhCH), 2.67 (dd, *J* = 13.4, 8.6 Hz, 1H, PhCH). ^13^C NMR (150 MHz, DMSO-*d*_6_) δ 171.31 (C=O), 167.14 (C=O), 167.06 (C=O), 162.55 (C=O), 158.97, 149.26, 140.22, 137.80, 135.94, 134.73, 132.59, 129.41 (2 × C), 129.07 (2 × C), 128.94 (2 × C), 128.64 (2 × C), 128.33 (2 × C), 126.93, 115.08 (2 × C), 104.77, 100.10, 55.86, 51.90, 50.08, 38.09, 37.75.ESI-MS: m/z 539.5 (M + 1)^+^, m/z 561.4 (M + 23)^+^. C_31_H_30_N_4_O_5_ [538.22].

##### (*S*)-4-fluoro-N-(1-(2-((1-((4-methoxyphenyl)(methyl)amino)-1-oxo-3-phenylpropan-2-yl)amino)-2-oxoethyl)-2-oxo-1,2-dihydropyridin-4-yl)benzamide (**TD-1b**)

Yellow solid, yield: 83%. m.p.: 124–126 ℃. ^1^H NMR (400 MHz, DMSO-*d*_6_) δ 10.31 (s, 1H, NH), 8.60 (d, *J* = 7.7 Hz, 1H, NH), 8.06–7.97 (m, 2H, Ph), 7.44–7.35 (m, 3H, Ph, Ph (2H), 2-Pyridone (1H)), 7.21 (d, *J* = 6.9 Hz, 3H, Ph (2H), 2-Pyridone (1H)), 7.07–6.98 (m, 2H, Ph), 6.95–6.87 (m, 5H, Ph), 6.56 (dd, *J* = 7.5, 2.4 Hz, 1H, 2-Pyridone), 4.57–4.36 (m, 3H, CH_2_, CH), 3.76 (s, 3H, OCH_3_), 3.09 (s, 3H, NCH_3_), 2.91–2.84 (m, 1H, PhCH), 2.67 (dd, *J* = 13.4, 8.6 Hz, 1H, PhCH). ^13^C NMR (151 MHz, DMSO-*d*_6_) δ 171.31 (C=O), 167.12 (C=O), 165.90 (C=O), 164.86 (d, ^1^*J_CF_* = 250.0 Hz), 162.52 (C=O), 158.97, 149.18, 140.25, 137.80, 135.94, 131.21, 131.15, 129.41 (2 × C), 129.07 (2 × C), 128.64 (2 × C), 126.92 (2 × C), 115.92 (d, ^2^*J_CF_* = 22.0 Hz, 2 × C), 115.08 (2 × C), 104.82, 100.06, 55.85, 51.90, 50.08, 38.09, 37.75.ESI-MS: m/z 557.3 (M + 1)^+^, m/z 579.4 (M + 23)^+^. C_31_H_29_FN_4_O_5_ [556.21].

##### (*S*)-3-fluoro-N-(1-(2-((1-((4-methoxyphenyl)(methyl)amino)-1-oxo-3-phenylpropan-2-yl)amino)-2-oxoethyl)-2-oxo-1,2-dihydropyridin-4-yl)benzamide (**TD-1c**)

Yellow solid, yield: 85%. m.p.: 118–120 ℃. ^1^H NMR (400 MHz, DMSO-*d*_6_) δ 10.36 (s, 1H, NH), 8.61 (d, *J* = 7.7 Hz, 1H, NH), 7.77 (dd, *J* = 14.3, 8.8 Hz, 2H, Ph), 7.61 (td, *J* = 8.0, 5.8 Hz, 1H, Ph), 7.49 (td, *J* = 8.5, 2.6 Hz, 1H, Ph), 7.41 (d, *J* = 7.5 Hz, 1H, 2-Pyridone), 7.26–7.18 (m, 3H, Ph (2H), 2-Pyridone (1H)), 7.07–6.99 (m, 2H, Ph), 6.96–6.88 (m, 5H, Ph), 6.57 (dd, *J* = 7.5, 2.4 Hz, 1H, 2-Pyridone (1H)), 4.59–4.38 (m, 3H, CH_2_, CH), 3.77 (s, 3H, OCH_3_), 3.10 (s, 3H, NCH_3_), 2.88 (dd, *J* = 13.7, 5.5 Hz, 1H, PhCH), 2.68 (dd, *J* = 13.4, 8.6 Hz, 1H, PhCH). ^13^C NMR (150 MHz, DMSO-*d*_6_) δ 171.31 (C=O), 167.10 (C=O), 165.66 (C=O), 162.49 (C=O), 162.37 (d, ^1^*J_CF_* = 244.8 Hz), 158.97, 149.00, 140.32, 137.80, 136.95 (d, ^3^*J_CF_* = 6.6 Hz), 135.94, 131.17 (d, ^2^*J_CF_* = 8.0 Hz), 129.41 (2 × C), 129.07, 128.64 (2 × C), 126.93, 124.58 (d, ^3^*J_CF_* = 2.8 Hz), 115.18 (d, ^2^*J_CF_* = 22.7 Hz), 115.08 (2 × C), 105.03, 100.03, 55.86, 51.90, 50.09, 38.09, 37.75. ESI-MS: m/z 557.3 (M + 1)^+^, m/z 579.4 (M + 23)^+^. C_31_H_29_FN_4_O_5_ [556.21].

##### (*S*)-2-fluoro-N-(1-(2-((1-((4-methoxyphenyl)(methyl)amino)-1-oxo-3-phenylpropan-2-yl)amino)-2-oxoethyl)-2-oxo-1,2-dihydropyridin-4-yl)benzamide (**TD-1d**)

Yellow solid, yield: 86%. m.p.: 96–99 °C. ^1^H NMR (400 MHz, DMSO-*d*_6_) δ 10.52 (s, 1H, NH), 8.59 (d, *J* = 7.7 Hz, 1H, NH), 7.66 (td, *J* = 7.4, 1.8 Hz, 1H, Ph), 7.60 (td, *J* = 7.6, 2.0 Hz, 1H, Ph), 7.42–7.36 (m, 2H, Ph), 7.33 (d, *J* = 7.6 Hz, 1H, 2-Pyridone), 7.20 (d, *J* = 6.8 Hz, 3H, Ph (2H), 2-Pyridone (1H)), 7.07–6.97 (m, 2H, Ph), 6.91 (dd, *J* = 7.4, 4.9 Hz, 4H, Ph), 6.84 (d, *J* = 2.3 Hz, 1H, Ph), 6.44 (dd, *J* = 7.5, 2.4 Hz, 1H, 2-Pyridone), 4.57–4.37 (m, 3H, CH_2_, CH), 3.76 (s, 3H, OCH_3_), 3.09 (s, 3H, NCH_3_), 2.91–2.84 (m, 1H, PhCH), 2.67 (dd, *J* = 13.4, 8.6 Hz, 1H, PhCH). ^13^C NMR (150 MHz, DMSO-*d*_6_) δ 171.30 (C=O), 167.08 (C=O), 164.28 (C=O), 162.49 (C=O), 159.36 (d, ^1^*J_CF_* = 249.4 Hz), 158.97, 148.73, 140.56, 137.79, 135.93, 133.54 (d, ^3^*J_CF_* = 8.4 Hz), 130.37 (d, ^4^*J_CF_* = 2.6 Hz), 129.41 (2 × C), 129.07 (2 × C), 128.64 (2 × C), 126.93, 125.11 (d, ^3^*J_CF_* = 3.4 Hz), 124.76 (d, ^2^*J_CF_* = 14.8 Hz), 116.71 (d, ^2^*J_CF_* = 21.6 Hz), 115.07 (2 × C), 104.53, 99.65, 55.85, 51.90, 50.12, 38.09, 37.75. ESI-MS: m/z 557.3 (M + 1)^+^, m/z 579.4 (M + 23)^+^. C_31_H_29_FN_4_O_5_ [556.21].

##### (*S*)-4-chloro-N-(1-(2-((1-((4-methoxyphenyl)(methyl)amino)-1-oxo-3-phenylpropan-2-yl)amino)-2-oxoethyl)-2-oxo-1,2-dihydropyridin-4-yl)benzamide (**TD-1e**)

Yellow solid, yield: 80%. m.p.: 121–125 °C. ^1^H NMR (400 MHz, DMSO-*d*_6_) δ 10.36 (s, 1H, NH), 8.61 (d, *J* = 7.8 Hz, 1H, NH), 7.95 (d, *J* = 8.6 Hz, 2H, Ph), 7.63 (d, *J* = 8.5 Hz, 2H, Ph), 7.40 (d, *J* = 7.5 Hz, 1H, 2-Pyridone), 7.21 (d, *J* = 6.8 Hz, 3H, Ph (2H), 2-Pyridone (1H)), 7.02 (d, *J* = 6.0 Hz, 2H, Ph), 6.94–6.87 (m, 5H, Ph), 6.56 (dd, *J* = 7.5, 2.4 Hz, 1H, 2-Pyridone), 4.59–4.36 (m, 3H, CH_2_, CH), 3.76 (s, 3H, OCH_3_), 3.09 (s, 3H, NCH_3_), 2.88 (dd, *J* = 13.5, 5.6 Hz, 1H, PhCH), 2.67 (dd, *J* = 13.4, 8.6 Hz, 1H, PhCH). ^13^C NMR (150 MHz, DMSO-*d*_6_) δ 171.30 (C=O), 167.11 (C=O), 165.94 (C=O), 162.50 (C=O), 158.97, 149.09, 140.29, 137.79, 137.50, 135.94, 133.40, 130.31 (2 × C), 129.41 (2 × C), 129.06 (2 × C), 129.03 (2 × C), 128.64 (2 × C), 126.92, 115.08 (2 × C), 104.93, 100.04, 55.86, 51.90, 50.09, 38.09, 37.75. ESI-MS: m/z 573.4 (M + 1)^+^, m/z 595.4 (M + 23)^+^. C_31_H_29_ClN_4_O_5_ [572.18].

##### (*S*)-4-bromo-N-(1-(2-((1-((4-methoxyphenyl)(methyl)amino)-1-oxo-3-phenylpropan-2-yl)amino)-2-oxoethyl)-2-oxo-1,2-dihydropyridin-4-yl)benzamide (**TD-1f**)

Yellow solid, yield: 88%. m.p.: 112–116 °C. ^1^H NMR (400 MHz, DMSO-*d*_6_) δ 10.36 (s, 1H, NH), 8.60 (d, *J* = 7.7 Hz, 1H, NH), 7.88 (d, *J* = 8.5 Hz, 2H, Ph), 7.77 (d, *J* = 8.5 Hz, 2H, Ph), 7.40 (d, *J* = 7.5 Hz, 1H, 2-Pyridone), 7.21 (d, *J* = 6.8 Hz, 3H, Ph (2H), 2-Pyridone (1H)), 7.07–6.98 (m, 2H, Ph), 6.95–6.87 (m, 5H, Ph), 6.56 (dd, *J* = 7.5, 2.4 Hz, 1H, 2-Pyridone), 4.57–4.37 (m, 3H, CH_2_, CH), 3.76 (s, 3H, OCH_3_), 3.09 (s, 3H, NCH_3_), 2.88 (dd, *J* = 13.5, 5.6 Hz, 1H, PhCH), 2.67 (dd, *J* = 13.5, 8.6 Hz, 1H, PhCH). ^13^C NMR (150 MHz, DMSO-*d*_6_) δ 171.30 (C=O), 167.10 (C=O), 166.07 (C=O), 162.50 (C=O), 158.97, 149.08, 140.29, 137.79, 135.93, 133.76, 132.16, 131.98 (2 × C), 131.74, 130.46 (2 × C), 129.41 (2 × C), 129.07, 128.64 (2 × C), 126.92, 115.08 (2 × C), 104.94, 100.04, 55.86, 51.90, 50.09, 38.09, 37.75. ESI-MS: m/z 617.4 (M + 1)^+^. C_31_H_29_BrN_4_O_5_ [616.13].

##### (*S*)-N-(1-(2-((1-((4-methoxyphenyl)(methyl)amino)-1-oxo-3-phenylpropan-2-yl)amino)-2-oxoethyl)-2-oxo-1,2-dihydropyridin-4-yl)-4-methylbenzamide (**TD-1g**)

Yellow solid, yield: 71%. m.p.: 118–120 °C. ^1^H NMR (400 MHz, DMSO-*d*_6_) δ 10.21 (s, 1H, NH), 8.60 (d, *J* = 7.7 Hz, 1H, NH), 7.84 (d, *J* = 8.0 Hz, 2H, Ph), 7.39 (d, *J* = 7.5 Hz, 1H, 2-Pyridone), 7.35 (d, *J* = 8.0 Hz, 2H, Ph), 7.21 (d, *J* = 6.8 Hz, 3H, Ph (2H), 2-Pyridone (1H)), 7.05–6.98 (m, 2H, Ph), 6.89 (s, 5H, Ph), 6.58 (dd, *J* = 7.5, 2.4 Hz, 1H, 2-Pyridone), 4.57–4.36 (m, 3H, CH_2_, CH), 3.76 (s, 3H, OCH_3_), 3.09 (s, 3H, NCH_3_), 2.88 (dd, *J* = 13.5, 5.6 Hz, 1H, PhCH), 2.67 (dd, *J* = 13.4, 8.6 Hz, 1H, PhCH), 2.39 (s, 3H, PhCH_3_). ^13^C NMR (150 MHz, DMSO-*d*_6_) δ 171.31 (C=O), 167.15 (C=O), 166.82 (C=O), 162.55 (C=O), 158.97, 149.33, 142.79, 140.16, 137.80, 135.93, 131.83, 129.47 (2 × C), 129.41 (2 × C), 129.07 (2 × C), 128.64 (2 × C), 128.39 (2 × C), 126.93, 115.07 (2 × C), 104.66, 100.12, 55.85, 51.90, 50.06, 38.09, 37.75, 21.50. ESI-MS: m/z 553.5 (M + 1)^+^, m/z 575.5 (M + 23)^+^. C_32_H_32_N_4_O_5_ [552.24].

##### (*S*)-4-methoxy-N-(1-(2-((1-((4-methoxyphenyl)(methyl)amino)-1-oxo-3-phenylpropan-2-yl)amino)-2-oxoethyl)-2-oxo-1,2-dihydropyridin-4-yl)benzamide (**TD-1h**)

Yellow solid, yield: 86%. m.p.: 109–112 °C. ^1^H NMR (400 MHz, DMSO-*d*_6_) δ 10.12 (s, 1H, NH), 8.57 (d, *J* = 7.7 Hz, 1H, NH), 7.93 (d, *J* = 8.7 Hz, 2H, Ph), 7.37 (d, *J* = 7.5 Hz, 1H, 2-Pyridone), 7.24–7.17 (m, 3H, Ph (2H), 2-Pyridone (1H)), 7.07 (d, *J* = 8.9 Hz, 2H, Ph), 7.01 (d, *J* = 8.3 Hz, 2H, Ph), 6.95–6.87 (m, 5H, Ph), 6.58 (dd, *J* = 7.5, 2.3 Hz, 1H, 2-Pyridone), 4.56–4.36 (m, 3H, CH_2_, CH), 3.84 (s, 3H, OCH_3_), 3.76 (s, 3H, OCH_3_), 3.09 (s, 3H, NCH_3_), 2.88 (dd, *J* = 13.7, 5.5 Hz, 1H, PhCH), 2.70–2.63 (m, 1H, PhCH). ^13^C NMR (150 MHz, DMSO-*d*_6_) δ 171.31 (C=O), 167.16 (C=O), 166.30 (C=O), 162.88 (C=O), 162.57, 158.97, 149.45, 140.09, 137.80, 135.94, 130.40 (2 × C), 129.41 (2 × C), 129.07 (2 × C), 128.64 (2 × C), 126.92, 126.71, 115.08 (2 × C), 114.21 (2 × C), 104.51, 100.15, 55.99, 55.85, 51.90, 50.05, 38.10, 37.75. ESI-MS: m/z 569.5 (M + 1)^+^, m/z 591.5 (M + 23)^+^. C_32_H_32_N_4_O_6_ [568.23].

##### methyl (*S*)-4-((1-(2-((1-((4-methoxyphenyl)(methyl)amino)-1-oxo-3-phenylpropan-2-yl)amino)-2-oxoethyl)-2-oxo-1,2-dihydropyridin-4-yl)carbamoyl)benzoate (**TD-1i**)

Yellow solid, yield: 75%. m.p.: 98–100 °C. ^1^H NMR (400 MHz, DMSO-*d*_6_) δ 10.48 (s, 1H, NH), 8.59 (d, *J* = 7.7 Hz, 1H, NH), 8.10 (d, *J* = 8.2 Hz, 2H, Ph), 8.04 (d, *J* = 9.2 Hz, 2H, Ph), 7.41 (d, *J* = 7.5 Hz, 1H, 2-Pyridone), 7.25–7.15 (m, 3H, Ph (2H), 2-Pyridone (1H)), 7.02 (d, *J* = 8.3 Hz, 2H, Ph), 6.95–6.87 (m, 5H, Ph), 6.57 (dd, *J* = 7.5, 2.3 Hz, 1H, 2-Pyridone), 4.58–4.38 (m, 3H, CH_2_, CH), 3.90 (s, 3H, COOCH_3_), 3.76 (s, 3H, OCH_3_), 3.09 (s, 3H, NCH_3_), 2.88 (dd, *J* = 13.6, 5.6 Hz, 1H, PhCH), 2.71–2.64 (m, 1H, PhCH). ^13^C NMR (150 MHz, DMSO-*d*_6_) δ 171.31 (C=O), 167.10 (C=O), 166.25 (C=O), 166.08 (C=O), 162.50 (C=O), 158.97, 149.00, 140.35, 137.79, 135.93, 130.02, 129.77, 129.68 (2 × C), 129.41 (2 × C), 129.07 (2 × C), 128.75 (2 × C), 128.64 (2 × C), 126.92, 115.08 (2 × C), 105.08, 100.03, 55.85, 52.90, 51.91, 50.11, 38.09, 37.75. ESI-MS: m/z 597.4 (M + 1)^+^. C_33_H_32_N_4_O_7_ [596.23].

##### (*S*)-4-cyano-N-(1-(2-((1-((4-methoxyphenyl)(methyl)amino)-1-oxo-3-phenylpropan-2-yl)amino)-2-oxoethyl)-2-oxo-1,2-dihydropyridin-4-yl)benzamide (**TD-1j**)

Yellow solid, yield: 83%. m.p.: 134–136 °C. ^1^H NMR (400 MHz, DMSO-*d*_6_) δ 10.51 (s, 1H, NH), 8.58 (d, *J* = 7.7 Hz, 1H, NH), 8.07 (d, *J* = 8.3 Hz, 2H, Ph), 8.03 (d, *J* = 8.2 Hz, 2H, Ph), 7.41 (d, *J* = 7.5 Hz, 1H, 2-Pyridone), 7.20 (d, *J* = 6.6 Hz, 3H, Ph (2H), 2-Pyridone (1H)), 7.01 (d, *J* = 8.2 Hz, 2H, Ph), 6.91 (dd, *J* = 5.7, 3.2 Hz, 5H, Ph), 6.55 (dd, *J* = 7.5, 2.3 Hz, 1H, 2-Pyridone), 4.59–4.37 (m, 3H, CH_2_, CH), 3.76 (s, 3H, OCH_3_), 3.09 (s, 3H, NCH_3_), 2.88 (dd, *J* = 13.6, 5.4 Hz, 1H, PhCH), 2.67 (dd, *J* = 13.4, 8.5 Hz, 1H, PhCH). ^13^C NMR (150 MHz, DMSO-*d*_6_) δ 171.30 (C=O), 167.07 (C=O), 165.73 (C=O), 162.46 (C=O), 158.97, 148.87, 140.43, 138.67, 137.79, 135.93, 132.98 (2 × C), 129.41 (2 × C), 129.19 (2 × C), 129.06 (2 × C), 128.64 (2 × C), 126.93, 115.08 (2 × C), 105.17, 105.10, 99.98, 99.90, 55.86, 51.91, 50.12, 38.09, 37.75. ESI-MS: m/z 564.5 (M + 1)^+^, m/z 586.4 (M + 23)^+^. C_32_H_29_N_5_O_5_ [563.22].

##### (*S*)-N-(1-(2-((1-((4-methoxyphenyl)(methyl)amino)-1-oxo-3-phenylpropan-2-yl)amino)-2-oxoethyl)-2-oxo-1,2-dihydropyridin-4-yl)-2-naphthamide (**TD-1k**)

Yellow solid, yield: 81%. m.p.: 113–117 °C. ^1^H NMR (400 MHz, DMSO-*d*_6_) δ 10.48 (s, 1H, NH), 8.59 (d, *J* = 8.0 Hz, 1H, NH), 8.57 (s, 1H, Naphthalene), 8.12–7.96 (m, 4H, Naphthalene), 7.70–7.60 (m, 2H, Naphthalene), 7.42 (d, *J* = 7.4 Hz, 1H, 2-Pyridone), 7.26–7.15 (m, 3H, Ph (2H), 2-Pyridone (1H)), 7.07–6.99 (m, 2H, Ph), 6.98 (d, *J* = 2.3 Hz, 1H, Ph), 6.95–6.88 (m, 4H, Ph), 6.63 (dd, *J* = 7.5, 2.3 Hz, 1H, 2-Pyridone), 4.59–4.38 (m, 3H), 3.76 (s, 3H), 3.09 (s, 3H), 2.88 (dd, *J* = 13.3, 5.8 Hz, 1H), 2.68 (dd, *J* = 13.5, 8.4 Hz, 1H). ^13^C NMR (150 MHz, DMSO-*d*_6_) δ 171.32 (C=O), 167.15 (C=O), 167.09 (C=O), 162.58 (C=O), 158.98, 149.32, 140.27, 137.80, 135.94, 134.98, 132.47, 132.00, 130.96, 129.52 (2 × C), 129.42 (2 × C), 129.08 (2 × C), 128.92, 128.65 (2 × C), 128.63 (2 × C), 128.18, 127.46, 126.93, 124.83, 115.09 (2 × C), 104.81, 100.13, 55.86, 51.91, 50.11, 38.10, 37.76. ESI-MS: m/z 589.4 (M + 1)^+^, m/z 611.4 (M + 23)^+^. C_35_H_32_N_4_O_5_ [588.24].

##### (*S*)-N-(1-(2-((1-((4-methoxyphenyl)(methyl)amino)-1-oxo-3-phenylpropan-2-yl)amino)-2-oxoethyl)-2-oxo-1,2-dihydropyridin-4-yl)-4-nitrobenzamide (**TD-1l**)

Yellow solid, yield: 81%. m.p.: 134–136 °C. ^1^H NMR (600 MHz, DMSO-*d*_6_) δ 10.61 (s, 1H, NH), 8.58 (d, *J* = 7.7 Hz, 1H, NH), 8.39–8.36 (m, 2H, Ph), 8.18–8.14 (m, 2H, Ph), 7.43 (d, *J* = 7.4 Hz, 1H, 2-Pyridone), 7.24–7.17 (m, 3H, Ph (2H), 2-Pyridone (1H)), 7.02 (s, 2H, Ph), 6.94–6.89 (m, 5H, Ph), 6.57 (dd, *J* = 7.5, 2.4 Hz, 1H, 2-Pyridone), 4.59–4.40 (m, 3H, CH_2_, CH), 3.76 (s, 3H, OCH_3_), 3.09 (s, 3H, NCH_3_), 2.88 (dd, *J* = 13.6, 5.6 Hz, 1H, PhCH), 2.68 (dd, *J* = 13.5, 8.5 Hz, 1H, PhCH). ^13^C NMR (150 MHz, DMSO-*d*_6_) δ 171.30 (C=O), 167.07 (C=O), 165.47 (C=O), 162.46 (C=O), 158.97, 149.95, 148.84, 140.45, 140.27, 137.79, 135.93, 129.92 (2 × C), 129.41 (2 × C), 129.07 (2 × C), 128.64 (2 × C), 126.93, 124.04 (2 × C), 115.08 (2 × C), 105.27, 99.99, 55.86, 51.91, 50.13, 38.09, 37.75. ESI-MS: m/z 584.4 (M + 1)^+^, m/z 606.4 (M + 23)^+^. C_31_H_29_N_5_O_7_ [583.21].

#### 5.1.7. Procedure for the Synthesis of **TD-1m**

**TD-1l** and 10% Pd/C (10% *w/w*) were dissolved in methanol (5 mL) and dichloromethane (5 mL), and the solution degassed and was stirred at room temperature for 2h under H_2_. The mixture was filtered and concentrated, and the resulting residue were purified by recrystallization or preparation thin layer chromatography to provide the target compounds **TD-1m**.

##### (*S*)-4-amino-N-(1-(2-((1-((4-methoxyphenyl)(methyl)amino)-1-oxo-3-phenylpropan-2-yl)amino)-2-oxoethyl)-2-oxo-1,2-dihydropyridin-4-yl)benzamide (**TD-1m**)

Yellow solid, yield: 65%. m.p.: 137–139 °C. ^1^H NMR (600 MHz, DMSO-*d*_6_) δ 9.79 (s, 1H, NH), 8.55 (d, *J* = 7.7 Hz, 1H, NH), 7.69 (d, *J* = 8.6 Hz, 2H, Ph), 7.33 (d, *J* = 7.5 Hz, 1H, 2-Pyridone), 7.24–7.17 (m, 3H, Ph (2H), 2-Pyridone (1H)), 7.01 (s, 2H, Ph), 6.94–6.88 (m, 5H, Ph), 6.61 (d, *J* = 8.6 Hz, 2H, Ph), 6.59 (dd, *J* = 7.5, 2.3 Hz, 1H, 2-Pyridone), 5.85 (s, 2H, NH_2_), 4.55–4.35 (m, 3H, CH_2_, CH), 3.76 (s, 3H, OCH_3_), 3.10 (s, 3H, NCH_3_), 2.87 (dd, *J* = 12.1, 4.4 Hz, 1H, PhCH), 2.67 (dd, *J* = 13.4, 8.4 Hz, 1H, PhCH). ^13^C NMR (150 MHz, DMSO-*d*_6_) δ 171.32 (C=O), 167.22 (C=O), 166.51 (C=O), 162.63 (C=O), 158.96, 153.19, 149.84, 139.85, 137.79, 135.93, 130.23 (2 × C), 129.41 (2 × C), 129.07, 128.64 (2 × C), 126.92, 120.68, 115.08 (2 × C), 113.04 (2 × C), 103.91, 100.23, 55.85, 51.88, 50.00, 38.09, 37.75. ESI-MS: m/z 554.5 (M + 1)^+^, m/z 576.4 (M + 23)^+^. C_31_H_31_N_5_O_5_ [553.23].

### 5.2. In Vitro Anti-HIV Assay in MT-4 Cells

Evaluation of the antiviral activity of the compounds against HIV in MT-4 cells was performed using the MTT assay as described below. Stock solutions (10 × final concentration) of test compounds were added in 25 µL volumes to two series of triplicate wells to allow for the simultaneous evaluation of their effects on mock- and HIV-infected cells at the beginning of each experiment. Serial five-fold dilutions of test compounds were made directly in flat-bottomed 96-well microtiter trays using a Biomek 3000 robot (Beckman Instruments, Fullerton, CA, USA). Untreated HIV- and mock-infected cell samples were included as controls. HIV stock (50 µL) at 100–300 CCID_50_ (50% cell culture infectious doses) or culture medium was added to either the infected or mock-infected wells of the microtiter tray. Mock-infected cells were used to evaluate the effects of the test compound on uninfected cells to assess the test compounds’ cytotoxicity. Exponentially growing MT-4 cells were centrifuged for 5 min at 220 g, and the supernatant was discarded. The MT-4 cells were resuspended at 6 × 10^5^ cells/mL, and 50 µL volumes were transferred to the microtiter tray wells. Five days after infection, the viability of mock-and HIV-infected cells was examined spectrophotometrically using the MTT assay. The MTT assay is based on the reduction of yellow colored 3-(4,5-dimethylthiazol-2-yl)-2,5-diphenyltetrazolium bromide (MTT) (Acros Organics, Belgium) by mitochondrial dehydrogenase activity in metabolically active cells to a blue-purple formazan that can be measured spectrophotometrically. The absorbances were read in an eight-channel computer-controlled photometer (Infinite M1000, Tecan, Swiss) at two wavelengths (540 and 690 nm). All data were calculated using the median absorbance value of three wells. The 50% cytotoxic concentration (CC_50_) was defined as the concentration of the test compound that reduced the absorbance (OD540) of the mock-infected control sample by 50%. The concentration achieving 50% protection against the cytopathic effect of the virus in infected cells was defined as the 50% effective concentration (EC_50_).

### 5.3. Solubility Studies

We weighed about 1mg of the compound, dissolved it with 1ml of methanol, and diluted it in gradient (twice each time) to obtain 10 concentrations. The peak area of the compound was tested by HPLC, and the peak area- concentration standard curve was drawn. In addition, about 1 mg, 0.6 mg and 0.3 mg of the compound were weighed and dissolved in 1 mL of pH 2, pH 7 and pH 7.4 phosphate buffer to obtain the saturated solution of the compound. HPLC was used to test the peak area under different pH, and the water solubility of the compound was obtained through the standard curve.

### 5.4. Molecular Docking Studies

All molecules were prepared using LigPrep (Maestro, Schrödinger, LLC, New York, NY, USA, 2021), and the hexameric HIV-1 CA (PDB ID: 5HGL) protein was used as a receptor for the docking calculations. The Glide application was used for all docking studies with default parameters. The docking results were visualized by PyMOL v2.3.0 (Schrödinger, New York, NY, USA).

### 5.5. Computational Assessment of Drug-like Properties, Metabolic Stability, and Toxicity

Drug-like properties, metabolic stability, and toxicity were assessed using the P450 and Derek Nexus module within Stardrop V7 [42] (Optibrium, Ltd., Cambridge, UK).

For metabolic stability evaluation, representative compounds were docked into the human CYP3A4 isoform (PDB ID 4D78). CYP3A4 was prepared using Autodock tools [49,50,51,52], where essential hydrogen atoms, Kollman united atom type charges, and solvation parameters were added. The grid box for the docking search was centered around the catalytic center of CYP3A4 [49,50,51]. Docking calculations were performed using AutoDock via DockingServer [52]. The docked poses were further evaluated with SeeSAR 12.1 (BioSolveIT Gmbh, Germany) utilizing the predictive binding affinity calculations using the HYdrogen bond and DEhydration (HYDE) energy scoring function [44,45,46]. The lower boundary of the affinity prediction for the poses with acceptable torsion angles and clash scores were used in combination with composite site lability (CSL), and the number of labile sites within the molecule to predict metabolic stability.

## Data Availability

Not applicable.

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
