# Peer review of "Design, Synthesis, and Mechanistic Study of 2-Pyridone-Bearing Phenylalanine Derivatives as Novel HIV Capsid Modulators"

_molecules, 2022, doi:10.3390/molecules27217640_

Round 1
Reviewer 1 Report
The manuscript by Dr Zhang and co-workers submitted for publication in “Molecules” describes a novel series of 2-Pyridone-bearing Phenylalanine Derivatives as HIV capsid inhibitors. These compounds were designed, synthesized, characterized, and evaluated as HIV capsid inhibitors. Most of the newly synthesized compounds displayed anti-HIV activity: FTC-2 is the most potent anti-HIV-1 compound, while TD-1a is more potent at blocking the replication of HIV-2. Overall, the obtained results in the current manuscript by the authors is very impressive and will be of interest to others in this field and hence I support for publication in “Molecules” after minor revisions.
Revision Request:
In Paragraph 1, page 2, please cite “Figure 2A” in the text.
In Paragraph 3.2., page 7, please remove the point after “same”.
Figure 5, please separate figure caption from text; which is the value of pKi? 0.3?
In Paragraph 5.1.4, page 12, please use the same name for “FTC-1” in the text. Sometimes the compound is indicated by “F-TC-1”.
Reviewer 2 Report
The compounds in this article submitted for publication in the journal Molecules contain new and beautiful activity results.
It can be published as it is, but a few additions would be nice.
1- page 4 can be redesigned.
2- 1H(13C)NMR spectra are not available for all molecules in supplementary files. It will be better for the molecule library if the missing ones are completed.
Reviewer 3 Report
This paper reported the design, synthesis, and biological evaluation of 2-pyridone-bearing phenylalanine derivatives as novel HIV capsid Inhibitors. In addition, the mechanistic study including binding and drug-like properties, and metabolic stability investigation were also carried out virtually. The potent compounds (FTC-2 and TD-1a) in anti-HIV cellular assay showed similar or less activity than the parent/control compound PF74. The virtual study displayed the good drug-like properties and metabolic stability of TD-1a compared with that of the parent/control compound PF74. This topic is interesting, and fits the scope of Molecules. The manuscript is well-organized, while key experiments are required to be supplemented to support the conclusions, and the key issues are required to be addressed before its publication on Molecules.
1. The protein target of the representative compounds is required to be validated with biochemical assay, such as fluorescence polarization assay, Alpha assay, or ITC/SPR/BLI binding assay.
2. Two more amides or its mimics were introduced to the target compounds which may lower the solubility of the target compound compared with the parent/control compound PF74. The authors are required to test the solubility of the representative compounds in this manuscript.
3. Methods about the computational assessment of drug-like properties and metabolic stability are missing.
4. The cellular anti-virus effects were evaluated through cytopathic effects (CPE) of virus in infected cells in this manuscript. The validation of antiviral activities (for representative compounds) with an orthometric assay (such as RT-qPCR assay to detect the copy of virus RNA) was suggested to be performed.
5. Per the discussion about the drug resistance problem of the current anti-HIV drugs in the manuscript, the representative new compounds were suggested to be assayed against the resistant strains of HIV virus, especially those with PF74 resistance.
Round 2
Reviewer 3 Report
After the authors’ revision according to my and others’ previous comments, the quality of this paper was significantly improved, and could reach the required quality standard for Molecules in my opinion. I suggest accepting it without further revision.